# Dynamic Linkage between Aging, Mechanizations and Carbon Emissions from Agricultural Production

**DOI:** 10.3390/ijerph19106191

**Published:** 2022-05-19

**Authors:** Lili Guo, Yuting Song, Shuang Zhao, Mengqian Tang, Yangli Guo, Mengying Su, Houjian Li

**Affiliations:** 1College of Economics, Sichuan Agricultural University, Chengdu 611130, China; 14453@sicau.edu.cn (L.G.); 201907364@stu.sicau.edu.cn (Y.S.); 201907285@stu.sicau.edu.cn (S.Z.); tangmengqian@stu.sicau.edu.cn (M.T.); 2School of Economics and Management, Southwest Jiaotong University, Chengdu 610031, China; guoyangli@my.swjtu.edu.cn; 3College of Economics, Guangxi Minzu University, Nanning 530006, China

**Keywords:** aging, agricultural mechanization, agricultural carbon emissions, PVAR

## Abstract

The trend of aging is intensifying and has become a prominent population phenomenon worldwide. The aging population has an important impact on carbon emissions, but at present, there is little research on its ecological consequences, especially the relationship with agricultural carbon emissions. For a long time, China has been dominated by a scattered small-scale peasant economy. Currently, the aging population also means that the agricultural labor force will gradually become scarce, and the agricultural production will face reform. This article is intended to find the long-term impact of aging and mechanization on agricultural carbon emissions and construct a more comprehensive policy framework for sustainable development, hoping to contribute to environmental and ecological protection. The research sample in this article is from 2000 to 2019, covering 30 provinces (cities, autonomous regions) in China. We adopted methods and models including Fully Modified General Least Squares (FMOLS), Dynamic General Least Squares (DOLS), Panel Vector Autoregression (PVAR) model, etc., and used the Granger causality test to determine the causal relationship between variables. Results show that aging is the Granger cause of agricultural carbon emissions and agricultural mechanization. Agricultural carbon emissions and agricultural mechanization have a bidirectional causal relationship. In the short term, agricultural mechanization and aging both have made a great contribution to carbon dioxide emissions from agricultural production. However, in the long term, the impact of aging on agricultural mechanization is significantly negative. Therefore, it is generally beneficial to improve the environmental problems of agricultural production. Our research focuses on the latest background of population trends and global climate issues and finally provides suggestions and a theoretical basis for the formulation of government agricultural policies according to the research conclusions.

## 1. Introduction

Carbon dioxide emissions and aging are the two major challenges facing the development of human society at present. Climate deterioration due to greenhouse gas emissions has become a major environmental problem worldwide [1,2], seriously threatening socio-economic development and human survival [3,4]. The United Nations Declaration on Sustainable Development Goals emphasized the importance of improving global environmental quality [5]. Global ecologists, policy makers and economists pay close attention to and actively advocate for emission reduction policies [6]. The international community has signed climate conventions such as the United Nations Framework Convention on Climate Change and the Paris Agreement to address the climate issues. In addition, aging is an important trend in changing the age structure of the world population, which has become a widespread social problem in developed countries [7] and developing countries [8]. The United Nations released a report “World Population Prospects 2019: Summary of Findings” on 17 June 2019 and pointed out that the proportion of the global population over 65 years old is currently about 9%, and by 2050, this proportion will reach 16%, which is the clearest manifestation of the aging of the world population. Aging has brought problems such as increasing social burden, weakening the function of family support for the aged, shortage of labor force, etc. At the same time, it has also seriously affected carbon emissions [9].

China’s carbon dioxide emission levels have led to it being recognized as the most polluted economy in the world [10], so it is urgent to take corresponding measures to reduce carbon emissions, especially to promote agricultural carbon emission reduction. China is an agricultural country with a large population, with 9% of the world’s cultivated land supporting about 21% of the world population [11]. A total of 17% of China’s greenhouse gas emissions come from agricultural production activities, so its role in aggravating climate warming cannot be ignored [12]. On the one hand, in pursuing the high growth of agricultural production, China has become the largest fertilizer producer and consumer globally. Chinese farmers use far more fertilizers per hectare than other countries [13]. In a short period of 35 years, from 1980 to 2015, China’s agricultural productivity has dramatically improved. The grain yield per unit area increased by 56%, but it was accompanied by a 225% increase in fertilizer input [14]. The input of pesticides, agricultural films, chemical fertilizers and other elements has led to severe soil pollution problems while increasing grain yield [15]. Their overuse has become the main cause of China’s agricultural non-point source pollution [16].

As an input of agricultural technology, an advanced representative of modern agricultural productivity and a key factor affecting agricultural carbon emissions, agricultural machinery plays a vital role in improving agricultural labor conditions, promoting agricultural economic output and adjusting agricultural industrial structure [17,18]. Agricultural machinery is mainly driven by fossil fuels, and in recent years, the rapidly growing energy consumption of agricultural mechanization (including irrigation motor pumps and tractors, etc.) has gradually become an essential source of carbon emissions [19]. The gas pollution produced by fossil energy consumption has a significant negative impact on the environment [20]. Compared with the early 1970s, the number of tractors used per 100 square kilometers of arable land in the early 21st century increased by nearly 5.5 times [21]; machinery mainly depends on burning fossil fuels such as kerosene and diesel. The emission coefficient of diesel is the highest among all energy sources, and its carbon emissions account for 15% of the total carbon emissions [22]. However, in the 17 years from 2000 to 2017, China’s agricultural energy consumption increased by nearly 50 million tons of standard coal (China Statistical Yearbook). Energy consumption increases the carbon dioxide discharged into the atmosphere [23]. Wiebe et al. [24] also clearly concluded that agricultural mechanization will exacerbate the carbon dioxide emissions of the agricultural sector through a survey of global carbon emissions from the agricultural sector. Jiang et al. [20] also pointed out that there were some shortcomings in energy saving and emission reduction of agricultural machinery, and the greenhouse gases emitted by agricultural machinery reduced the eco-environmental performance. Nevertheless, it is undeniable that the popularization of agricultural mechanization has significantly improved the efficiency of agricultural production [25]. In order to give consideration to the ecological environment and agricultural production efficiency, agricultural production urgently needs to organically integrate agricultural mechanization with a low-carbon economy through scientific and technological innovation, which can effectively help protect the ecological environment and play a role in energy conservation and emission reduction [26].

Since the 1990s, China’s aging process has been accelerating. At present, China has already stepped into an aging society, with the largest number of the elderly (aged 60 and above) in the world, and the aging process in China will continue to accelerate in the future [27]. In China’s rural areas, farmers generally have small landholdings and large families, so their agricultural income is relatively low. With the acceleration of urbanization, a large number of rural people move to cities to pursue higher-paid non-agricultural employment activities [28,29], leading to the increasingly serious aging problem in rural China. Statistics show that from 1982 to 2019, the aging rate in countryside China increased by 11.24%, from 4.56% to 15.8% [30]. It is also necessary to explore the relationship between aging and agricultural carbon emissions. The direct relationship between them is the consumption pattern of the elderly. Scholars have different views on whether aging will increase or decrease carbon emissions. Some scholars believe that the elderly spend a higher proportion of their income on energy-intensive products such as housing, food and health care [31]. At the same time, aging accelerates the miniaturization of families, leading to more energy consumption and carbon dioxide emissions [32]. However, most studies still believe that aging directly reduces carbon emissions, which are negatively related to carbon emissions, and can improve the environment [33,34]. An important reason for the above conclusion is that the living habits of the elderly are more low-carbon and environmentally friendly than that of the younger generation [33], and their overall consumption demand is smaller [35], they spend more time indoors, rarely use transportation and reduce the energy consumption of private cars [36]. According to China’s energy demand table, the annual demand of the elderly is 0.7 times that of the young, and the aging suppresses food demand and reduces calories [37]. Hassan and Salim [38] studied the influence of aging on carbon dioxide emissions in 25 high-income countries and proved that for every 1% increase in the elderly population, the per capita carbon dioxide emissions would decrease by 1.55%.

In addition, aging also affects agricultural carbon emissions. On the one hand, in the context of rural labor shortages, the “high-input, high-output” agricultural production model has been widely adopted in many regions, expanding the use of chemical fertilizers [39]. Due to their low level of education and environmental awareness, the elderly in rural areas do not care about agricultural non-point source pollution during agricultural production [40], which increases agricultural carbon emissions. On the other hand, the physiological decline in the elderly chose traditional primitive farming methods or reduced farming. Research shows that farming will affect methane (CH_4_) emission, and no-till or minimal tillage is regarded as a means to reduce carbon dioxide emission and promote soil carbon sequestration, reducing leaching or surface runoff [41,42]. The change in population age also determines the direction of land utilization transformation and agricultural development [43]. Under the influence of aging and urbanization, young people migrate to cities, and there are some problems in the agricultural sector, such as the shortage of agricultural labor, the increase in cost and the weakening of the family’s economic development ability, resulting in the formerly cultivated land being idle [44]. Therefore, farmers will adjust their land-use decisions to adapt to the impact of aging on agricultural production [45]. In this way, farmers can still get the greatest economic benefits when the labor supply decreases [46]. One way of adjustment is to abandon the land and make it non-agricultural [47] or let the land be idle. Abandoned farmland has a positive role in mitigating climate change, replacing agricultural cultivation with natural regeneration, which can cause better carbon sequestration [48]. The second method is to change the types of crops planted, choose the types of crops that are labor-saving and easy to grow, use less labor [49], reduce the planting scale of food crops and increase the area of cash crops. The last way is to lease the land. The shortage of agricultural labor force caused by aging can promote the land transaction between the families lacking labor force and those with surplus labor force [50]. According to the research of Deininger and Jin [51], in China, the rural land leasing market can improve land utilization rate by allocating land to those with high agricultural potential and then achieving large-scale operation. Moreover, aging has led to the fragmentation and decentralization of small farmers’ management, making it challenging to form large-scale production, which will seriously hinder mechanization, reduce the applicability and universality of large-scale agricultural machinery [52], weaken the use of agricultural machinery and reduce mechanical energy consumption and CO_2_ emissions.

However, the long-term aging of the population will eventually affect food security and challenge sustainable development [53]. The population aging will encourage farmers to make up for the shortage of labor by increasing the input of machinery and improving the level of agricultural modernization [43]. Agricultural mechanization is more widely used as an essential substitute for labor input [17]. Although it leads to more carbon emissions, it greatly improves agricultural production efficiency [18,54]. Therefore, efforts to reduce environmental pollution caused by agricultural machinery may be the right way for sustainable development. Research widely has shown that the critical factor in improving energy efficiency and controlling carbon dioxide emissions growth is technological progress [55]. In the early stage, due to less investment in scientific research and low technical level, it was not conducive to reducing energy consumption and carbon dioxide emissions. In the later stages of mechanization, the government encouraged renewable energy use [56], continuously increased the investment of capital and technical personnel, and made remarkable progress in energy-saving technology. By reducing the energy intensity of machinery [57] and adopting comprehensive agricultural technology [25], China has gradually promoted agricultural industrialization. While improving the efficiency of agricultural machinery, it has also gradually reduced the amount of agricultural machinery. Nowadays, introducing clean energy such as biogas and solar energy as the main power of agricultural machines [58] while improving mechanization efficiency and promoting technological innovation for ecological protection [26] can effectively reduce agricultural carbon emissions.

Related research on the two themes of carbon emissions and population aging has become a recent research hotspot for scholars and achieved fruitful results. Previous studies have laid a good foundation for our study, with important inspiration and reference significance. Existing literature has extensively studied the sources and influencing factors of carbon emissions, such as the impact of urbanization, industrial structure, energy consumption and land use on carbon emissions. However, most literature selects a single core variable to explore the linear relationship between this variable and carbon emissions. Compared with traditional research, this paper innovatively brings aging, agricultural mechanization and agricultural carbon emissions into a unified research system, examines their long-term equilibrium relationship, discusses the impact of aging and agricultural mechanization on carbon emissions and demonstrates the impact and transmission mechanism of aging on agricultural carbon emissions through agricultural mechanization, as well as the dynamic relationship between them. The above is also the biggest contribution of this paper. In addition, this paper has the following two contributions: 1. Using systematic and rich empirical methods, including a set of testing frameworks (cross-section correlation test, unit root test and cointegration test), and using Fully Modified General Least Squares (FMOLS) and Dynamic General Least Squares (DOLS) to make an empirical analysis of the long-term impact of variables, the estimated results are accurate and reliable. Using the Panel Vector Autoregression (PVAR) model, the endogenous problem and lag effect of variables are fully considered, which provides an effective testing method for studying the dynamic relationship. 2. In the past, scholars mainly used county-level data, survey data and data from several designated provinces and cities to study the impact of land change or aging on agricultural production. The research level is relatively low. However, this paper examines the situation of 30 provincial levels in China, with a wider research scope and more convincing results.

## 2. Materials and Methods

### 2.1. Data Interpretation and Index Construction

#### 2.1.1. Total Agricultural Carbon Emissions

According to Altieri and Nicholls [59], the diversity of agricultural carbon emission sources is caused by the diversity and complexity of agricultural production. According to FAO (Food and Agriculture Organization of the United Nations) data, agricultural land emissions, rice cultivation, agricultural waste disposal, animal enteric fermentation and animal manure management constitute the primary GHG (Greenhouse Gas) emissions from agricultural production activities in China. According to Cui et al. [60], the main sources of agricultural carbon emissions include pesticides, agricultural plastic films, fertilizers and agricultural activities. Among the many factors affecting agricultural carbon emissions, we mainly focus on the factors of production inputs and animal activities. Therefore, the carbon emission sources we selected mainly include fertilizer, pesticides, agricultural plastic film, agricultural planting, agricultural irrigation, agricultural electricity, agricultural diesel oil and the weight of pigs, cattle and sheep, estimating each province’s total agricultural carbon emissions. The fertilizer is a compound fertilizer containing nitrogen, phosphorus and potassium. The data sources of the above carbon sources are shown in Table 1. The table presents the main variable’s name, each variable’s unit and the data source.

In this paper, the emission coefficient method is used to calculate agricultural carbon emissions. In this method, agricultural carbon emissions are equal to carbon source consumption multiplied by the corresponding carbon emission factor. The formula is as follows:$$E=\sum E_{i}=\sum T_{i}\cdot \delta_{i}$$

In the above formula, $T_{i}$ is the carbon source usage, $\delta_{i}$ is the carbon emission coefficient of each carbon source (*i* represents the type of carbon source, *i* = 1, 2, …, 10), and *E* represents all agricultural carbon emissions.

Table 2 shows each carbon source’s emission factors and main reference sources. In the carbon emission factor unit in the second column of the table, the first unit represents the unit of carbon dioxide emissions: kg and the latter unit represents the measurement unit of each carbon source. Figure 1 shows the annual carbon emissions from agricultural production in China from 2000 to 2019.

#### 2.1.2. The Index of Population Aging and Mechanization

In this article, the population aging index is constructed by the ratio of the elderly over 65 years old to the total rural population. The mechanization index can be expressed by mechanization intensity and measured by agricultural machinery’s total power/cultivated land area. The data sources of aging are China Rural Statistical Yearbook and China Population, Employment Statistical Yearbook and China Statistical Yearbook. The data source of mechanization is China Rural Statistical Yearbook.

The carbon emission index used in this article is average carbon emission per land = sum of carbon emissions/cultivated land area.

#### 2.1.3. Descriptive Statistics Analysis

Table 3 reveals descriptive statistics of each variable. The data interval of total carbon emissions and total mechanical strength fluctuates obviously, which indicates that there are regional differences in agricultural mechanization use and carbon emissions in 30 provinces (cities, autonomous regions) across the country, which are related to the agricultural planting area and planting methods of each province (city, autonomous region). The national total carbon emissions and total mechanical strength are relatively large, so the average value is at a high level. The fluctuation of the aging interval is slight, and aging is common in all regions. In order to reasonably control the influence of heteroscedasticity, these data were taken logarithmically, and the standard deviation of the three variables after logarithm was large, showing significant differences.

### 2.2. Cross-Sectional Dependence Tests

A cross-section correlation test is an important part of panel inspection. Before determining whether a panel data set can carry out a series of tests, it is most important to conduct a cross-section test [63]. The significance of testing the original hypothesis of the cross-section is the weak cross-section correlation between variables. If the final result is significant, the original hypothesis will be rejected, which indicates that there is a strong cross-sectional correlation between aging, agricultural mechanization and agricultural carbon emissions.

To test the cross-section correlation, Breusch and Pagan put forward the Breusch–Pagan_LM_ test in 1980. In order to solve the shortcomings of the Breusch–Pagan_LM_ test, Pesaran [64] improved it and put forward Pesaran_CD_ and Pesaran_LM_ tests. The formulas of the above three cross-section inspection methods are as follows:(1)$${\mathrm{Breush}–\mathrm{Pagan}}_{\mathrm{LM}}=\overset{N-1}{\underset{i=1}{\sum }}•\overset{N}{\underset{j=i+1}{\sum }}T_{ij}\mu_{ij}^{2}\to \chi^{2}\left(\frac{N\left(N-1\right)}{2}\right)\;$$
(2)$${\mathrm{Pesaran}}_{\mathrm{LM}}=\sqrt{\frac{1}{N\left(N-1\right)}}\overset{N-1}{\underset{i=1}{\sum }}•\overset{N}{\underset{j=i+1}{\sum }}\left(T_{ij}\mu_{ij}^{2}-1\right)\to N\left(0,1\right)\;$$
(3)$${\mathrm{Pesaran}}_{\mathrm{CD}}=\sqrt{\frac{2}{N\left(N-1\right)}}\overset{N-1}{\underset{i=1}{\sum }}•\overset{N}{\underset{j=i+1}{\sum }}T_{ij}\mu_{ij}^{2}\to N\left(0,1\right)\;$$
(4)$$\mu_{ij}=\mu_{ij}=\frac{\sum_{t-1}^{T}\varepsilon_{ij}\varepsilon_{ji}}{{\left(\sum_{t-1}^{T}\varepsilon_{ij}^{2}\;\right)}^{\frac{1}{2}}{\left(\sum_{t-1}^{T}\varepsilon_{jt}^{2}\;\right)}^{1/2}}$$

When the number of samples *N* and the time series *T* are small, the Formula (1) can be used. The data set with a large sample and dynamic time variation is suitable for Equation (2), while Equation (3) is suitable for large sample and fixed time.

In Equation (4), $\mu_{ij}^{2}$ is the correlation coefficient of residual error, $\varepsilon_{ij}$ and $\varepsilon_{ji}$ refer to standard errors.

### 2.3. Unit Root Test

#### 2.3.1. IPS

IPS test [65] points out that the $\rho_{i}$ difference of some individuals can also cause the instability of the whole panel data. IPS test overcomes the defect of the LLC test and allows different individuals in the panel to have different $\rho_{i}$, which is a relaxation of the hypothesis. The relaxation of IPS homogeneity requirements is more in line with the characteristics of economic data and acknowledges that the whole data is stable and some individual data are unstable.

The form of the IPS inspection model is as follows:$$\Delta y_{i,t}=\alpha_{t}+\rho_{i}y_{i,t-1}+\varepsilon_{i,t},\;i=1,2,\ldots ,N,\;t=1,2,\ldots ,T$$

In the IPS test, the original hypothesis and alternative hypothesis are as follows:$$H_{0}:\;\rho_{i}=0,\forall_{i}\in N$$



$$H_{1}:\{\begin{array}{c} \rho_{i}<0,i=1,2,\ldots ,N_{1}\\ \rho_{i}=0,i=N_{1}+1,N_{2}+2,\ldots ,T \end{array}$$



Based on the modified DF-*t* statistic, the original hypothesis test statistic:$$\Gamma_{t}=\sqrt{N}\left[t_{NT}\left(p\right)-\alpha_{NT}\right]/\sqrt{b_{NT}}\to N\left(0,1\right)$$
$$\alpha_{NT}=\frac{1}{N}\overset{i=N}{\underset{i=1}{\sum }}E[t_{NT}\left(p,0\right)]$$
$$b_{NT}=\frac{1}{N}\overset{i=N}{\underset{i=1}{\sum }}var[t_{NT}\left(p,0\right)]$$

In the above formula, when the lag period is *p*, $t_{NT}\left(p\right)$ is the ADF-*t* statistic of *N* departments, and $var\left[t_{NT}\left(p,0\right)\right]$ and $E\left[t_{NT}\left(p,0\right)\right]$ represent the variance and mean of the ADF-*t* statistic of *N* departments with a lag period of *p*, respectively.

Under the original hypothesis, *T* → ∞, *N* → ∞, or *N*/*T* → *k*, *k* is a finite normal number. Statistics converge to normal distribution function.

#### 2.3.2. LLC

LLC (Levin–Lin–Chu) [66] test is suitable for the same root and long panel data (*T* > *n*). The test principle adopts the ADF test, but it uses standard proxy variables that exclude the influence of autocorrelation and deterministic items. LLC assumes sequence correlation from the beginning and uses the ADF test to test whether there is a unit root. The ADF test model equation is as follows:$$\Delta y_{it}=\rho y_{i\;t-1}+\overset{p_{i}}{\underset{L=1}{\sum }}\theta_{iL}\Delta y_{i\;t-L}+\alpha_{mi}d_{mt}+\varepsilon_{it},\;m=1,2,3,$$

Assuming that all individuals in LLC have homogeneity under the original and alternative assumptions, the following are the original and alternative assumptions:$$H_{0}:\;\rho_{1\;}=\rho_{2}\;=\ldots =\rho_{N}=0,$$
$$H_{1}:\;\rho_{1\;}=\rho_{2}\;=\ldots =\rho_{N}<0.$$

Under the above null hypothesis, $\hat{\rho }$ and $se\left(\hat{\rho }\right)$ are calculated using the combined data, and $t_{\rho }=\frac{\hat{\rho }}{se(\hat{\rho })}$ and modified $\frac{t_{\rho }-N\tilde{T}\hat{S_{N}}\hat{\sigma_{\tilde{\varepsilon }}}{-}^{2}STD(\hat{\rho })\mu_{m\tilde{T}}^{*}}{\sigma_{m\bar{T}}^{*}}$.

Statistics and $t_{\rho }=\frac{\hat{\rho }}{\mathrm{STD}(\hat{\rho })}$ are calculated by least squares, and the statistics contained in the modified $t_{\rho }^{*}=\frac{t_{\rho }-N\tilde{T}\hat{S_{N}}\hat{\sigma_{\tilde{\varepsilon }}}{-}^{2}STD(\hat{\rho })\mu_{m\tilde{T}}^{*}}{\sigma_{m\bar{T}}^{*}}$, and $t_{\rho }^{*}$ are as follows:$$\hat{\rho }=\frac{\sum_{i=1}^{N}\sum_{t=2+p_{i}}^{T}{\tilde{v}}_{it-1}{\tilde{e}}_{it}}{\sum_{i=1}^{N}\sum_{t=2+p_{i}}^{T}{\tilde{v}}_{it-1}^{2}}$$
$$\mathrm{STD}\hat{\rho }={\tilde{\sigma }}_{\tilde{\varepsilon }}{\left(\sum_{i=1}^{N}\sum_{t=2+p_{i}}^{T}{\tilde{v}}_{it-1}^{2}\right)}^{-1/2},$$
$${\hat{\rho }}_{\tilde{\varepsilon }}^{2}\frac{1}{N\tilde{T}}=\overset{N}{\underset{i=1}{\sum }}\sum_{t=2+p_{i}}^{T}{\left({\tilde{e}}_{it}-\hat{\rho }{\tilde{v}}_{it-1}\right)}^{2}.$$
where $\tilde{T}=T-\bar{P}-1$, $\bar{P}=\frac{1}{N}\overset{N}{\underset{i=1}{\sum }}P_{i}$.

$t_{\rho }^{*}$ converges to the standard normal distribution.

#### 2.3.3. ADF

The high-order autoregressive process may produce time series in the actual unit root test, and the random error term may not be white noise series. Therefore, in order to ensure random error terms’ noise characteristics in the DF test, Dicky and Fuller proposed an enhanced DF test, which formed the enhanced Dickey–Fuller test.

For theoretical and practical reasons, the following three regression models are commonly used for the ADF test:(5)$$\Delta y_{t}=\bar{\omega }y_{t-1}+\overset{k}{\underset{i=1}{\sum }}\beta_{i}\Delta y_{\;t-i}+\varepsilon_{t}$$
(6)$$\Delta y_{t}=\alpha +\bar{\omega }y_{t-1}+\overset{k}{\underset{i=1}{\sum }}\beta_{i}\Delta y_{\;t-i}+\varepsilon_{t}$$
(7)$$\Delta y_{t}=\alpha +\delta t+\bar{\omega }y_{t-1}+\overset{k}{\underset{i=1}{\sum }}\beta_{i}\Delta y_{\;t-i}+\varepsilon_{t}\;\;$$

The original assumptions in the three models are $H_{0}:\delta =0$, and there is a unit root. In model (7), *T* represents the trend of time series changing with time, which is a time variable, and *α* is a constant term. The inspection starts with model (7), then model (6), and finally model (5). The inspection will not stop until the original hypothesis is rejected by the inspection, which means that the original series has no unit root. It is a stationary series.

#### 2.3.4. PP

The unit root test method of PP is put forward for the existence of sequence correlation of disturbance items. Phillips and Perron revised the ADF test nonparametric in 1988 and put forward the Phillips–Perron test statistic. This test statistic obeys the limit distribution of the corresponding ADF test statistic and applies to the stationarity test of heteroscedasticity.

The steps of PP inspection are as follows:

① Estimating the regression model by the least square method to obtain the parameter estimation and residual sequence;

② Calculate the sample autocovariance of residual sequence:$${\hat{\gamma }}_{j}=T^{-1}\overset{T}{\underset{t=j+1}{\sum }}{\hat{u}}_{t}{\hat{u}}_{t-j},j=0,1,2,\ldots .$$

The estimated value of λ=σφ(1) is as follows:$${\hat{\gamma }}^{2}={\hat{\gamma }}_{0}+2\overset{q}{\underset{j-1}{\sum }}\left[1-\frac{j}{q+1}\right].{\hat{\gamma }}_{j}$$

The size of the value of *q* is determined according to the actual situation. If after a certain order (such as after the nth order), the contribution of ${\hat{\gamma }}_{j}$ to ${\hat{\gamma }}^{2}$ is negligible, then *q* takes *n*.

③ Calculate the standard deviation ${\hat{\sigma }}_{\hat{\rho }}$ of the parameter estimator $\hat{\rho }$ and the estimated variance $s^{2}=\frac{1}{T-2}\sum {\hat{u}}_{t}^{2}$ of residual $u_{t}$.

④ Substitute the calculation result in ③ into the expression of $Z_{\rho }$ or $Z_{t}$ statistic to get the statistic value.

### 2.4. Panel Cointegration Test

If the unit root process is found, it is necessary to continue the panel cointegration test to see if there is a long-term equilibrium cointegration relationship between endogenous variables. Kao [67] and Kao and Chiang [68] used the generalized DF and ADF tests to test panel cointegration. The initial assumption of this method is that there is no cointegration relationship between variables, and the residual of static panel regression is used to test the build statistics. The inspection process is divided into the following two stages:

The first stage: set each section to have different intercept terms and the same coefficient:$$y_{it}=\alpha_{i}+\delta_{i}t+x_{it}^{’}\beta +u_{it}\;\;\;$$
where $\alpha_{i}$ is different and β is the same, and is set.

In the second stage, the residual sequence ${\hat{u}}_{it}$ in the first stage is tested by unit root. Under the original hypothes is H_0_:ρ = 1, construct the following statistics:$${\mathrm{ADF}}_{\rho }=\frac{\sqrt{NT}\left(\hat{\rho }-1\right)+3\sqrt{N}{\hat{\sigma }}_{v}^{2}/{\tilde{\sigma }}_{v}^{2}}{\sqrt{3+36{\hat{\sigma }}_{v}^{4}/\left(5{\tilde{\sigma }}_{v}^{4}\right)}}$$
$${\mathrm{ADF}}_{t}=\frac{t_{\rho }+\sqrt{6N}{\hat{\sigma }}_{v}^{2}/\left(2{\tilde{\sigma }}_{v}\right)}{\sqrt{\frac{{\tilde{\sigma }}_{v}}{2{\tilde{\sigma }}_{v}}+3{\hat{\sigma }}_{v}^{2}/\left(10{\tilde{\sigma }}_{v}^{2}\right)}}$$

### 2.5. Causality Test

Granger [69] initiated the analysis of the causality of time series data. On this basis, Dumitrescu Hurlin [70] extended it. Granger causality means that when forecasting *Y*, the effect of forecasting with the past information containing variables *X* and *Y* are better than that of forecasting only with the past information of *Y*. This study adopts the causal test method proposed by Dumitrescu Hurlin to determine the directional causal relationship between variables. It includes directional causality in three ways: bidirectional causality, unidirectional causality, and neutral causality.

Granger test is completed by constrained F test, which is expressed as follows:$$Y_{t}=\overset{m}{\underset{i=1}{\sum }}\alpha_{i}X_{t-i}+\overset{m}{\underset{i=1}{\sum }}\beta_{i}Y_{t-i}+\mu_{1t}\;H_{0}:\;\alpha_{1}=\alpha_{2}=\cdots =\alpha_{m}=0,\;F=\frac{(SSR_{r}-SSR_{ur})/m}{SSR_{ur}/\left(n-k\right)}.$$

If F > F*_α_*(*m*, *n* − *k*), reject the original hypothesis: *x* is not the Granger cause of *Y*.
$$X_{t}=\overset{m}{\underset{i=1}{\sum }}\lambda_{i}Y_{t-i}+\overset{m}{\underset{i=1}{\sum }}\delta_{i}X_{t-i}+\mu_{2t}\;H_{0}:\;\lambda_{1}=\lambda_{2}=\cdots =\lambda_{m}=0,\;F=\frac{(SSR_{r}-SSR_{ur})/m}{SSR_{ur}/\left(n-k\right)}.$$

If F < F*_α_*(*m*, *n* − *k*), reject the original hypothesis that *y* is not the Granger cause of *X*.

Among them, if the regression model contains constant terms, the degree of freedom of F test *k* = 2*m* + 1, and if the constant terms are not included, *k* = 2*m*.

### 2.6. Autoregressive Distributed Lag

The autoregressive distributed lag (ARDL) method Pesaran [71] proposed can be used for long-term estimation of aging, agricultural mechanization and agricultural carbon emissions. Compared with the standard cointegration test, this method can test the long-term relationship between variables without single integration of variables of the same order.

The structure of a typical $\mathrm{ARDL}\left(p,q_{1},q_{2},\ldots q_{k}\right)$ model is as follows:$$\emptyset \left(L,P\right)y_{t}=\overset{k}{\underset{i=1}{\sum }}\beta_{i}(L,q_{i})x_{it}+\delta w_{t}+\mu_{t}$$
where
$$\emptyset \left(L,P\right)=1-\emptyset_{1}L-\emptyset_{2}L^{2}-\cdots -\emptyset_{p}L^{p}$$
$$\beta_{i}\left(L,q_{i}\right)=1-\beta_{i1}L-\beta_{i2}L^{2}-\cdots -\beta_{iqi}L^{qi}$$

*p* indicates the lag order of *Y*, *q_i_* indicates the lag order of *i*-th independent variable *x_i_*, *i* = 1, 2, …, *k. L* is the lag operator. The following formula can define it:$$Ly_{t}=y_{t-1}$$

### 2.7. FMOLS and DOLS

In this paper, after using the cointegration test and ARDL to test the long-term influence between test variables, we use FMOLS developed by Phillips and Hansen [72] and DOLS developed by Kao and Chiang [68] to run more robust tests. Compared with other regression methods, OLS and DOLS have the advantage that they can solve the problems of sequence correlation and endogenous explanatory variables in the study of long-term relationships between variables. Both of them are group average estimation methods between dimensions.

The panel FMOLS estimator *β* is given by:$$\beta_{NT}^{*}=N^{-1}\overset{N}{\underset{i=1}{\sum }}{\left(\overset{T}{\underset{i=1}{\sum }}{\left(X_{it}-{\bar{X}}_{i}\right)}^{2}\right)}^{-1}(\overset{T}{\underset{i=1}{\sum }}\left(X_{it}-{\bar{X}}_{i}\right)Y_{it}^{*}-T_{\hat{\tau_{i}}}$$
$$Y_{it}^{*}=\left(Y_{it}-\bar{Y_{i}}\right)-\frac{\hat{L_{21i}}}{\hat{L_{22i}}}\Delta X_{it},\;\hat{\tau_{i}}\equiv \hat{\Gamma_{21i}}+\hat{{Ω}_{21i}^{0}}-\frac{\hat{L_{21i}}}{\hat{L_{22i}}}\left(\hat{\Gamma_{22i}}+\hat{{Ω}_{22i}^{0}}\right)$$

The DOLS is written as follows:$$Y_{it}=\alpha_{i}+\beta_{i}X_{it}+\overset{jl}{\underset{j=-ji}{\sum }}\theta_{ij}\Delta X_{it-j}+\varepsilon_{it}^{*}$$
where the *β* is given by:$$\beta_{dols}^{*}=N^{-1}\overset{N}{\underset{i=1}{\sum }}{\left(\sum_{t=1}^{T}Z_{it}Z_{it}^{i}\right)}^{-1}\left(\overset{T}{\underset{t=1}{\sum }}Z_{it}Y_{it}^{*}\right)$$
where $Z_{it}=\left(X_{it}-\bar{X_{i}},\Delta X_{it-j},\ldots ,\Delta X_{it+k}\right)$ is 2(*K* + 1) vector of regressors.

### 2.8. PVAR

Panel autoregression, PVAR for short, was first proposed by Holtz Eakin et al. [73]. This model is based on a multivariate system equation, which converts all variables into endogenous variables for processing, and considers the effects of all variables’ lag terms. Based on inheriting the advantages of the VAR model, compared with the long time sequence requirement of the traditional VAR model, the PVAR model has the characteristics of a large cross-section and short time sequence. At the same time, the model can effectively solve the problem of individual heterogeneity by using panel data and fully considering the individual and time effects.

The PVAR formula is as follows:$$Y_{it}=\alpha_{0}+\overset{n}{\underset{j=1}{\sum }}\alpha_{j}Y_{i,i-j}+\beta_{i}+\gamma_{i}+\varepsilon_{it}$$

In the above formula, $Y_{it}$ refers to the core variable in this paper: agricultural carbon emission, aging and agricultural mechanization; i refers to the sample; t refers to the year; $\alpha_{0}$ refers to the intercept term; j refers to the lag order; $\alpha_{j}$ refers to the parameter matrix of lag order j; $\beta_{i}$ refers to the individual fixed effect; $\gamma_{i}$ refers to the individual time effect; and $\varepsilon_{it}$ refers to the random disturbance term that obeys the normal distribution.

## 3. Results

### 3.1. Cross-Sectional Dependence and Unit Root Tests Results

The results of the cross-section correlation test are shown in Table 4. We can conclude that the original hypothesis is rejected, and the alternative hypothesis is accepted at 1% significance level. There is a cross-sectional correlation.

Empirically, we usually use a unit root test to analyze the stationarity of panel data to check whether the data process is stable. This process is to avoid “false regression” and ensure the validity of the estimation results. In order to ensure the reliability and robustness of the test results, this paper uses the LLC method for homogeneous panel hypothesis and IPS, ADF-Fisher and PP-Fisher method for heterogeneous panel hypothesis to test each variable’s stationarity.

Stationarity estimates for the three variables are reported in Table 5. The results obtained by the four methods of unit root test are basically the same. In terms of level value, only lnmachine in the LLC test rejects the original hypothesis with the unit root at a 1% significance level, and other variables cannot reject the original hypothesis. Therefore, we further test their first-order differences. The results show that at the 1% significance level, the null hypothesis with unit root is rejected for each variable, and they have the same order stability.

### 3.2. Panel Cointegration Test Results

From the cointegration test results shown in Table 6, we can find that the *p*-value is 0.0000, far less than 0.01. Therefore, we can reject the original hypothesis at a 1% significance level. That is, the alternative hypothesis can be supported, which means that there is a cointegration relationship between endogenous variables. We can draw the conclusion that there is a long-term balanced causal relationship between aging, mechanization and agricultural carbon emission reduction, which enables us to study how aging and mechanization affect agricultural CO_2_ emissions.

### 3.3. Results of DOLS and FMOLS

Table 7 shows the estimated results of the FMOLS and DOLS panel models. In the two regressions, the coefficient of mechanization is positive, but the coefficient of aging is negative, both of which are significant at a 1% significance level. It shows that mechanization will increase agricultural carbon emissions, which is consistent with previous studies [20,63]. On the contrary, aging can reduce emissions. According to the estimates of FMOLS and DOLS, when the mechanization index increases by 1%, agricultural carbon emissions will increase by 0.60% and 0.70%, respectively. When the proportion of the elderly population increases by 1%, agricultural carbon emissions will decrease by 0.76% and 0.89%, respectively, which shows that aging can directly promote the reduction of agricultural carbon emissions.

### 3.4. Results of ARDL

After ordinary regression analysis, in order to study the long-term effects of aging and mechanization on agricultural carbon emissions, we use the ARDL model for long-term and short-term regression. ARDL (1 1, 1, 1, 1) is selected for estimation in this paper because lower lag order can obtain more reliable results [74]. The results in Table 8 show that in the long run, mechanization will increase carbon emissions while aging will reduce carbon emissions, which is similar to the estimated results obtained by the FMOLS and DOLS methods as mentioned above. At the significance level of 1%, mechanization has a short-term positive impact on agricultural carbon emissions, while aging has no obvious impact on carbon emissions in the short term. This shows that it is a long-term process for the mechanism of aging to reduce carbon emissions, and the use of mechanization can immediately increase carbon emissions, so reducing the use of mechanization in the short term can effectively inhibit the increase of agricultural carbon emissions.

### 3.5. PVAR Robustness Test

In this study, we built a PVAR model of aging, agricultural mechanization and agricultural carbon emissions and got the optimal index when the lag period was 15. In order to ensure the effectiveness of applying the Granger causality test, impulse response and variance decomposition based on the PVAR model, we test the robustness of the PVAR model, that is, whether the modulus of the eigenvalue of the dynamic matrix is less than 1 (in the unit circle). As shown in Figure 2, the ideal result is obtained, and all points of the inverse root of the AR characteristic polynomial are in the circle. The PVAR model constructed in this paper is robust.

### 3.6. Causality Test Results

Cointegration implies Granger causality, but there is no guarantee that the direction of causality between variables can be identified. Therefore, in order to identify whether there is Granger causality between variables, we further carried out the Granger causality test. Table 9 shows the paired causality test results between agricultural carbon emissions, aging and agricultural mechanization. The original hypothesis shows that there is no Granger causality between them. We find that there is a two-way causal relationship between mechanization and carbon emissions at the significance level of 5%. At the significance level of 10%, there is a two-way causal relationship between aging and carbon emissions. In addition, aging is the one-way Granger cause of mechanization at the significant level of 5%.

### 3.7. Impulse Response and Variance Decomposition Analysis Results

Impulse response function and variance decomposition tools can be used to test the relationship and influence degree between aging, agricultural mechanization and agricultural carbon emission variables. By observing the impulse response image, we can get how the variables in the PVAR model react to each impact with time. Variance decomposition decomposes the fluctuation of endogenous variables into mutually explanatory components. By analyzing the contribution rate of various structural shocks to endogenous variables, the relative importance can be evaluated.

In this paper, the reaction time is set to 15 years. In Figure 3, the abscissa indicates the lag length of the impact, the ordinate indicates the response degree of endogenous variables to the impact, and the dashed lines on both sides of the solid line are the values of plus or minus two standard deviations of the impulse response function, indicating the possible range of impulse response. The response of carbon emissions to its impact reached the maximum value in the first period, gradually decreased with time, and reached the minimum value in the 15th period. Carbon emissions did not respond immediately to mechanization. The first reaction was 0, then began to fluctuate in a small range near the 0 axis. The fourth reaction decreased to 0 and then remained in an adverse reaction state, which tended to be stable for a long time, indicating that mechanization did cause carbon emissions in a short time. Similarly, the first reaction of carbon emissions to aging is 0, and then it is stable in a negative reaction state, which shows that aging can reduce carbon emissions and be effective for a long time. The response of mechanization to carbon emissions is the largest in the first period, and with the passage of time, it shows a rapid downward trend, gradually approaching zero, which shows that carbon emissions do not constitute a permanent state of mechanization. The response to mechanization increased in the first two periods, reached the maximum value (about 0.9) in the third period, and then decreased with a relatively gentle trend. Except for zero in the first stage, the response of mechanization to aging is negative, which shows that aging effectively reduces the use of mechanization. The response of aging to carbon emissions is negative within 1–15 and slowly decreases. The response of aging to mechanization is also negative. After the second period, it decreases to the minimum and then keeps stable at this value, and the impact effect is basically unchanged. The response of aging to aging is maintained at a positive and stable level, with a relatively rapid decline from the first to the second stage and a relatively gentle decline in the later stage.

In order to further evaluate the influence degree of the disturbance term of the model on the impact of endogenous variables and the contribution degree of different structural impacts during the change of each variable, we further decomposed the variance of the PVAR model (see Table 10). Let us take Issue 5, Issue 10 and Issue 15 for analysis.

We can conclude that the variance contribution rate of agricultural CO_2_ emissions comes almost from itself. Compared with the 5th and 10th periods, the 15th period decreases, but it still reaches 97.97%. The variance contribution rate of aging and mechanization is 1.14% and 0.88% when it lags behind 15 periods. The variance contribution rate of mechanization mainly comes from agricultural carbon emissions, followed by itself, and finally from aging. The variance contribution rate of mechanization to itself is gradually increasing, reaching 45.05% when it lags for 15 periods, while the contribution rates of agricultural carbon emissions and aging to mechanization are 49.38% and 5.57%, respectively. The variance contribution rate of aging is mainly from itself. When it lags for 15 periods, the contribution rates of itself, agricultural carbon emissions and mechanization to aging are 96.23%, 3.15% and 0.62%, respectively.

## 4. Discussion

In the first step of this study, we first tested the cross-sectional correlation of panel data to ensure that there is a correlation among variables and that we can study their dynamic relationship. On this basis, we continued the following four types of unit root tests, including the IPS test, LLC test, Fisher ADF test and Fisher PP test, to analyze the stationarity of panel data and avoid “false regression”. The test results show that after the first-order difference, all variables have passed the single root test, and all variables are a single integer of the first order, which can be tested by the progressive cointegration test. We use the Kao cointegration test to test that although some of the original core variables are unstable, in the long run, there is a cointegration relationship among all variables so that the PVAR model can be established, which lays the foundation for the follow-up research on the dynamic relationship among them. The corresponding DOLS, FMOLS and ARDL regression results in Table 7 and Table 8 demonstrated the different impacts of mechanization and aging on agricultural carbon emissions and obtained their respective impact coefficients. Then, we focus on the Granger causality test, impulse response and variance decomposition. Granger causality test verifies whether there is causality among variables. Impulse response analysis is used to measure the impact track of a standard difference of random disturbance on the current and future values of other variables, intuitively depict the dynamic interaction and effect among variables, and judge the time lag relationship among variables from the dynamic response. Variance decomposition can more accurately measure the degree of interaction and explanation among variables, and their results are shown in Table 9, Figure 3 and Table 10.

## 5. Conclusions and Recommendation

### 5.1. Conclusions

This paper uses the panel data on China’s aging, agricultural mechanization and carbon emissions from 2000 to 2019, puts the three core variables into the same system, constructs a research framework to explore the dynamic relationship, and supplements the existing literature. At first, the panel data were verified, including cross-sectional correlation, unit root test and cointegration test, which proved that there was a cross-sectional relationship between aging, mechanization and carbon emissions. Then, the PVAR model was established by using FMOLS and DOLS to regress the long-term equilibrium relationship, and variance decomposition and impulse response analysis were carried out. The results show that mechanization increases agricultural carbon emissions, while aging reduces carbon footprint, which is beneficial to realizing agricultural carbon reduction. In the short term, aging will weaken the use of machinery. However, in the long term, agricultural production efficiency and economies of scale will still be improved by agricultural machinery, and technological innovation will be an effective way to reduce mechanized energy consumption and curb carbon emissions.

### 5.2. Policy Suggestions

According to the above research conclusions, the following policy implications were drawn: first of all, correctly understand and pay attention to the aging phenomenon, give full play to the positive role of the elderly groups in carbon reduction, attach importance to the favorable changes brought by population changes to society, accelerate the cultivation of new professional farmers, and improve farmers’ environmental awareness in agricultural production. Secondly, strengthen cooperation with developed countries, actively introduce advanced agricultural production modes and technologies, at the same time, increase financial input and subsidies for scientific research, promote the progress and renewal of agricultural technologies, improve the efficiency of agricultural machinery, and encourage the adoption of renewable energy and clean energy to be used as equipment fuel to reduce the input of pesticides and fertilizers, and reduce the agricultural sector’s carbon dioxide emissions and energy consumption. Finally, actively promote the agglomeration of agricultural land, recycle abandoned land and fragmented land, implement unified management and guide scale business operation. Improve the application of the Internet of Things and artificial intelligence in the agricultural field, promote smart agriculture, improve agricultural green total factor productivity, and continue to transform into green agriculture.

### 5.3. Future Research Directions

The research topic of this paper focuses on the frontier and fits the latest background related to population and environmental issues, which are also important in the world in the future. It is the first time that aging, agricultural mechanization and agricultural carbon emissions are linked. Based on the data of 30 provinces (cities) in China, it is demonstrated on a larger scale, and a conclusion is drawn, which has excellent novelty and crucial practical significance. Future research can be further studied in two aspects: 1. Extending the research area and scope to the outside world, the actual situation of different countries may be different, so it is necessary to conduct more studies in other countries and draw more universal conclusions so that all countries in the world can discuss coping strategies. 2. The DID model can be further used to compare the differences between the control group and the treatment group before and after implementing the carbon reduction policy, to test the effect of the policy and improve the scientific and practical effect of the policy.

## Figures and Tables

**Figure 1 ijerph-19-06191-f001:**
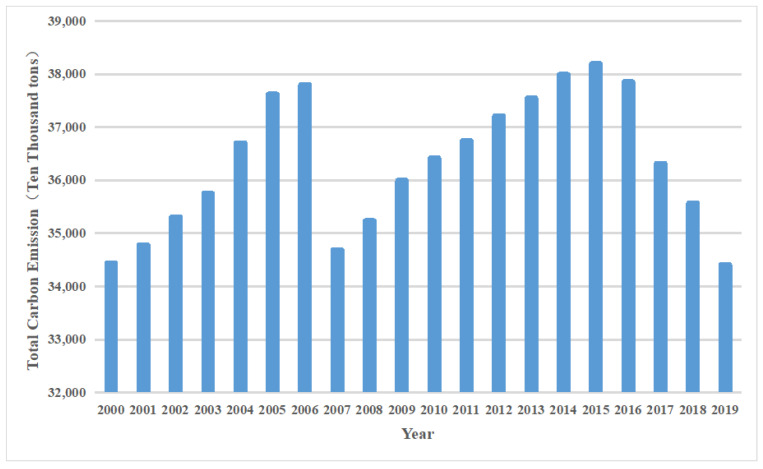
Carbon emissions from agricultural production from 2000 to 2019 in China.

**Figure 2 ijerph-19-06191-f002:**
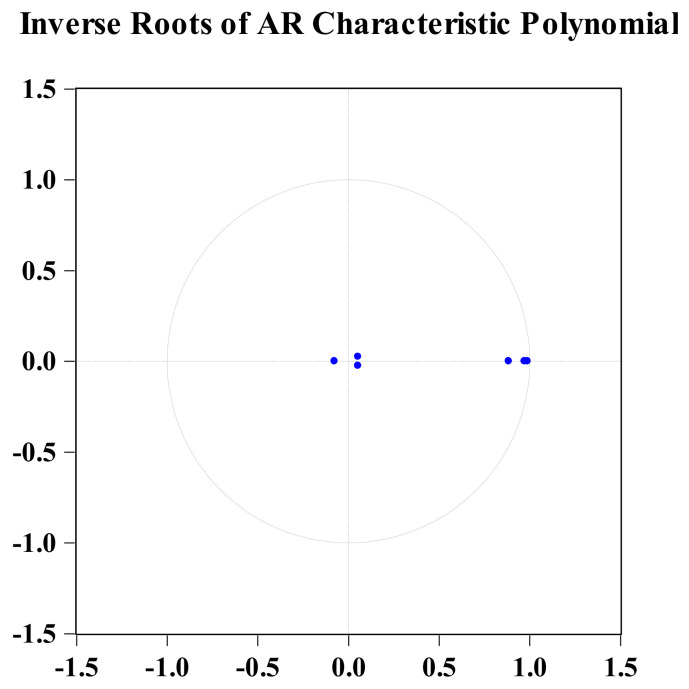
The inverse roots of the AR characteristic polynomial. Note: Blue bullet represents unit root, all blue bullets are inside the unit circle, which means that all unit roots are inside the unit circle.

**Figure 3 ijerph-19-06191-f003:**
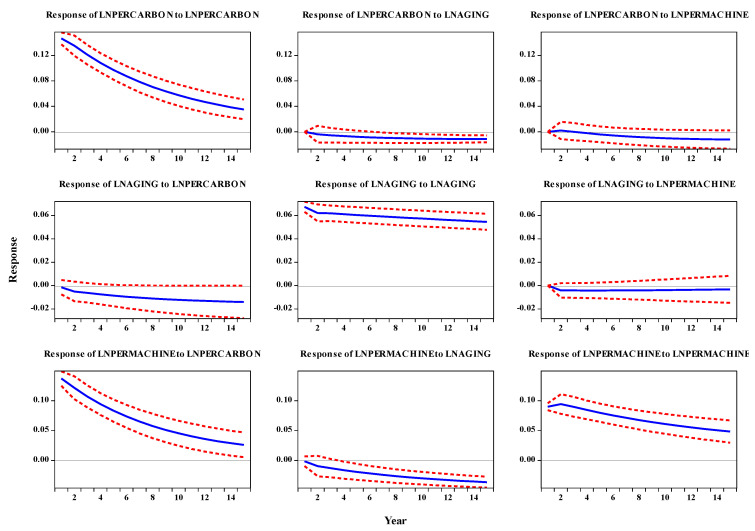
Impulse response diagram of three variables. Note: The horizontal axis represents time, and the vertical axis reflects the impact. The solid blue line in the middle represents the impulse response function, and the red dotted lines on both sides represent the upper and lower lines of the 95% confidence interval, respectively.

**Table 1 ijerph-19-06191-t001:** Data sources.

Variables	Unit	Data Sources
The pure amount of agricultural chemical fertilizer application	kg	
Pesticides consumption	kg	
Agricultural plastic films consumption	kg	
The total sown area of agriculture	hm^2^	
The effective irrigated area of agriculture	hm^2^	China Rural Statistical Yearbook
Agricultural power	kwh	
Agricultural diesel oil consumption	kg	
Pig	kg	
Cattle	kg	
Sheep	kg	

**Table 2 ijerph-19-06191-t002:** Carbon emission factors and reference sources.

Carbon Source	Carbon Emission Coefficient	Refer to the Main Source
Fertilizer	0.8956 kg/kg	Oak Ridge National Laboratory [61]
Pesticide	4.9341 kg/kg	Oak Ridge National Laboratory
Agricultural Plastic films	5.18 kg/kg	Institute of Resource, Ecosystem and Environment of Agriculture, Nanjing Agricultural University
Agricultural Power	CO_2_: 0.7921 t·MWh^−1^	Ministry of Ecology and Environment
Agricultural Cultivation	312.6 kg/hm^2^	College of Biological Sciences, China Agricultural University
Agricultural Irrigation	25 kg/hm^2^	[61,62]
Agricultural Diesel oil	0.5927 kg/kg	Intergovernmental Panel on Climate Change IPCC
Pig	34.0910 kg/(each year)	Intergovernmental Panel on Climate Change IPCC
Cattle	415.91 kg/(each year)	Intergovernmental Panel on Climate Change IPCC
Sheep	35.1819 kg/(each year)	Intergovernmental Panel on Climate Change IPCC

**Table 3 ijerph-19-06191-t003:** Descriptive statistics for primary variables.

Variable	Mean	Std. Dev.	Min	Max
Fertilizer	175.131	137.603	6.2	716.1
Pesticide	5.266	4.224	0.14	17.35
Agricultural plastic films	6.955	6.33	0.06	34.35
Agricultural diesel oil	63.827	64.681	1.8	487
Agricultural cultivation	5327.101	3588.664	88.6	14,783.4
Agricultural irrigation	2004.289	1515.2	109.24	6177.59
Agricultural electricity	212.195	337.304	1.5	1949.1
Cattle	357.893	293.474	1.2	1496.2
Pig	1515.626	1282.969	13.2	5757
Sheep	973.076	1200.783	11	6111.9
Total carbon emission	630.554	414.268	18.776	1996.382
Total mechanical strength	2809.358	2683.924	93.97	13,353.02
Machine	7.229	5.318	0.64	48.648
Aging	0.096	0.021	0.048	0.164
Cultivated area	207.21	119.33	1	412
Percarbon	1.866	2.315	0.357	22.985
Lnpercarbon	0.433	0.505	−1.03	3.135
Lnaging	−2.369	0.22	−3.044	−1.809
Lnmachine	1.779	0.639	−0.447	3.885

**Table 4 ijerph-19-06191-t004:** Cross-sectional dependence test results.

Test	Statistic	Prob.
Breusch–Pagan_LM_	2331.903	0.0000 ***
Pesaran scaled_LM_	64.31103	0.0000 ***
Pesaran_CD_	22.04594	0.0000 ***

Note: *** Significant at 1% level.

**Table 5 ijerph-19-06191-t005:** Panel unit root tests results.

Variables	Level		First-Difference	
	Intercept	Intercept and Trend	Intercept	Intercept and Trend
LLC test				
Lnpercarbon	0.6898	0.9999	0.0000 ***	0.0000 ***
Lnaging	0.9687	0.0845 *	0.0000 ***	0.0000 ***
Lnmachine	0.0000 ***	0.9992	0.0000 ***	0.0000 ***
Im, Pesaran, and Shin test				
Lnpercarbon	0.9691	1.0000	0.0000 ***	0.0000 ***
Lnaging	1.0000	0.2029	0.0000 ***	0.0000 ***
Lnmachine	0.4025	1.0000	0.0000 ***	0.0000 ***
ADF-Fisher Chi-square test				
Lnpercarbon	0.9037	0.9564	0.0000 ***	0.0000 ***
Lnaging	0.9988	0.1046	0.0000 ***	0.0000 ***
Lnmachine	0.5432	1.0000	0.0000 ***	0.0000 ***
PP-Fisher Chi-square test				
Lnpercarbon	0.9539	1.0000	0.0000 ***	0.0000 ***
Lnaging	1.0000	0.4615	0.0000 ***	0.0000 ***
Lnmachine	0.2209	1.0000	0.0000 ***	0.0000 ***

Note: *** Significant at 1% level, and * Significant at 10% level.

**Table 6 ijerph-19-06191-t006:** The results of Kao’s residual panel cointegration test (ADF).

	Null Hypothesis	*t*-Statistics	Probability
ADF	No co-integration	−7.227329	0.0000 ***

Note: *** Significant at 1% level.

**Table 7 ijerph-19-06191-t007:** Benchmark results.

Variables	Coefficient	S. E.	*t*-Statistic	Prob.
FMOLS				
LNMACHINE	0.6047	0.0346	17.4567	0.0000 ***
LNAGING	−0.7558	0.0894	−8.4559	0.0000 ***
DOLS				
LNMACHINE	0.6971	0.0438	15.9096	0.0000 ***
LNAGING	−0.8893	0.1280	−6.9468	0.0000 ***

Note: *** Significant at 1% level.

**Table 8 ijerph-19-06191-t008:** The results of ARDL.

Variable	Coefficient	Std. Error	*t*-Statistic	Prob.*
	Long Run Equation			
LNMACHINE	0.4446	0.0291	15.3011	0.0000 ***
LNAGING	−0.3578	0.0597	−5.9928	0.0000 ***
	Short Run Equation			
COINTEQ01	−0.1431	0.0488	−2.9342	0.0035 ***
D(LNMACHINE)	0.3612	0.0725	4.9842	0.0000 ***
D(LNAGING)	0.0623	0.0392	1.5900	0.1126
C	−0.1905	0.0618	−3.0842	0.0022 ***

Note: * represents a significant level, *** Significant at 1% level.

**Table 9 ijerph-19-06191-t009:** Pairwise Granger causality tests.

Null Hypothesis:	F-Statistic	Prob.
LNMACHINE does not Granger Cause LNPERCARBON	4.4489	0.0000 ***
LNPERCARBON does not Granger Cause LNMACHINE	2.4414	0.0140 **
LNAGING does not Granger Cause LNPERCARBON	8.2804	0.0000 ***
LNPERCARBON does not Granger Cause LNAGING	1.7365	0.0898 *
LNAGING does not Granger Cause LNMACHINE	2.3651	0.0177 **
LNMACHINE does not Granger Cause LNAGING	1.2932	0.2467

Note: *** Significant at 1% level, ** Significant at 5% level, and * Significant at 10% level.

**Table 10 ijerph-19-06191-t010:** The impulse response and variance decomposition results.

**Variance Decomposition of Lnpercarbon:**				
**Period**	**S. E.**	**Lnpercarbon**	**Lnmachine**	**Lnaging**
1	0.1466	100.0000	0.0000	0.0000
2	0.1995	99.9509	0.0115	0.0375
3	0.2334	99.9096	0.0085	0.0820
4	0.2574	99.8501	0.0133	0.1366
5	0.2753	99.7666	0.0323	0.2012
6	0.2890	99.6587	0.0666	0.2747
7	0.2996	99.5280	0.1159	0.3561
8	0.3081	99.3768	0.1792	0.4441
9	0.3149	99.2077	0.2549	0.5374
10	0.3204	99.0235	0.3416	0.6348
11	0.3249	98.8272	0.4377	0.7351
12	0.3287	98.6214	0.5415	0.8371
13	0.3318	98.4086	0.6515	0.9399
14	0.3345	98.1914	0.7662	1.0424
15	0.3367	97.9717	0.8843	1.1440
**Variance Decomposition of Lnmachine:**				
1	0.1640	69.9510	30.0490	0.0000
2	0.2253	66.2910	33.5832	0.1257
3	0.2654	64.0385	35.6795	0.2820
4	0.2946	62.2579	37.2564	0.4857
5	0.3169	60.6915	38.5699	0.7386
6	0.3347	59.2530	39.7064	1.0406
7	0.3492	57.9074	40.7023	1.3904
8	0.3613	56.6378	41.5764	1.7858
9	0.3715	55.4353	42.3405	2.2242
10	0.3804	54.2942	43.0035	2.7023
11	0.3882	53.2103	43.5729	3.2167
12	0.3952	52.1803	44.0557	3.7640
13	0.4015	51.2011	44.4584	4.3405
14	0.4072	50.2698	44.7875	4.9427
15	0.4125	49.3837	45.0491	5.5672
**Variance Decomposition of Lnaging:**				
1	0.0673	0.0474	0.0283	99.9243
2	0.0919	0.3305	0.3262	99.3433
3	0.1110	0.5452	0.4449	99.0099
4	0.1270	0.7659	0.5104	98.7237
5	0.1410	0.9931	0.5512	98.4557
6	0.1535	1.2242	0.5785	98.1973
7	0.1649	1.4566	0.5972	97.9463
8	0.1753	1.6877	0.6098	97.7025
9	0.1851	1.9156	0.6181	97.4664
10	0.1942	2.1388	0.6229	97.2384
11	0.2027	2.3561	0.6250	97.0189
12	0.2108	2.5667	0.6250	96.8083
13	0.2185	2.7702	0.6232	96.6066
14	0.2258	2.9661	0.6200	96.4139
15	0.2327	3.1544	0.6156	96.2300

## Data Availability

Data supporting the conclusions of this article are included within the article. The dataset presented in this study are available on request from the corresponding author.

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
