# Peer review of "Dynamic Linkage between Aging, Mechanizations and Carbon Emissions from Agricultural Production"

_ijerph, 2022, doi:10.3390/ijerph19106191_

Round 1
Reviewer 1 Report
Dear author,
The paper is good, you can improve:
Line 201-202: maybe is good to be reformulated. "The table presents the main variable’s name, each variable’s unit, and the data source."
Line 2: table 2. These values are for one year? Because I don’t see be specified.
It would be to use an impersonal expression in scientific articles, even if it is your study. For example: we draw the…/ it was drawn (line 557).
Author Response
First of all, we would like to thank Reviewer 1 for reading our article and for your valuable comments. Below is our response to all comments made.
Point 1:Line 201-202: maybe is good to be reformulated. "The table presents the main variable s name, each variable's unit, and the data source.’’
Response 1: The new expression suggested by the reviewers is indeed more concise and academic. We have put "the leftmost column is the name of the main variable, the middle column is the unit of each variable, And the right column is the data source correlating to each variable. " is replaced by "The table presents the main variable s name, each variable's unit, and the data source.”
Point 2:Line 2: table 2. These values are for one year? Because I don't see be specified. It would be to use an impersonal expression in scientific articles, even if it is your study. For example: we draw the.../it was drawn (line 557).
Response 2:
â‘ The values in Table 2 are not for one year, but obtained from the annual data of China's agricultural production carbon emissions from 2000 to 2019. For this reason, we have added a new data graph below Table 2 in the article, which can be seen more clearly.
â‘¡We sincerely accept your suggestion. “we draw the...” has been changed to “it was drawn” in the original text.
Reviewer 2 Report
The paper under review considered the dynamic linkage between aging, mechanizations and carbon emissions from agricultural production. In this paper, the authors explore the long-term effects of aging and mechanization on agricultural carbon emissions. By building a more comprehensive policy framework for sustainable development, the authors hope to contribute to environmental and ecological protection.
The content of this article is meaningful and the research method is correct. The author's discussion is interesting. The author presents a series of suggested policies that have potential application.
There are some typos in this article. For example,
Page 1, line 18,
“…on agricultural carbon emissions, construct …” -> “…on agricultural carbon emissions, and construct …”
Page 5, Line 219
“Mechanization index can be expressed by…” -> “The mechanization index can be expressed by…”
The reference of each curve in Figure 2 is not clear, so the author is suggested to add legends.
The authors conclude that mechanization can lead to an increase in agricultural carbon emissions, while aging can help achieve agricultural carbon reduction by reducing the carbon footprint. However, the author's discussion in this aspect is not sufficient, so it is suggested that the author provide detailed discussion.
The author's policy recommendations include reducing the amount of agricultural machinery used. Although reducing the number of agricultural machinery used can reduce carbon emissions, it will lead to a decrease in production efficiency. It is suggested that the author give discussion and explanation.
Carbon emissions have been studied in the literature. The authors are advised to compare this paper with the existing studies.
The paper can be considered for publication after revision.
Author Response
First of all, we would like to thank Reviewer 2 for reading our article and for your valuable comments. Below is our response to all comments made.
Point 1: Page 1, line 18,
“...on agricultural carbon emissions, construct ... -> ... on agricultural carbon emissions, and construct …”
Response 1: I'm very sorry, it's our negligence. We have revised the article according to your opinion.
Point 2: Page 5, Line 219
“Mechanization index can be expressed by…" -> “The mechanization index can be expressed by...”
Response 2: We have revised the article according to your opinion.
Point 3: The reference of each curve in Figure 2 is not clear; so the author is suggested to add legends.
Response 3: According to your suggestion, we have added the description of the relevant curve at the bottom of Figure 2, which is as follows: The solid blue line in the middle represents the impulse response function, and the red dotted lines on both sides represent the upper and lower lines of the 95% confidence interval, respectively.
Point 4: The authors conclude that mechanization can lead to an increase in agricultural carbon emissions, while aging can help achieve agricultural carbon reduction by reducing the carbon footprint. However, the author's discussion in this aspect is not sufficient, so it is suggested that the author provide detailed discussion.
Response 4: Thank you for your suggestion. In the introduction, we have made a fuller discussion on the increase of agricultural carbon emissions caused by mechanization and the reduction of agricultural carbon emissions caused by aging.
Point 5: The author's policy recommendations include reducing the amount of agricultural machinery used. Although reducing the number of agricultural machinery used can reduce carbon emissions, it will lead to a decrease in production efficiency. It is suggested that the author give discussion and explanation.
Response 5: Through the reading of the literature, it is true that our previous consideration is lacking. Although reducing the number of agricultural machinery used reduces carbon emissions, it will lead to lower production efficiency and food supply problems. The use of agricultural mechanization is the future trend. Therefore, the focus of our policy recommendations is ultimately to increase the use of clean energy and the improvement of science and technology to better improve the agricultural environment, which has been revised in the original text.
Point 6: Carbon emissions have been studied in the literature. The authors are advised to compare this paper with the existing studies.
Response 6:We sincerely accept your suggestion. Compare and summarize the research on carbon emissions in the existing literature with the research in this paper, and modify it as follows: Existing literatures have extensively studied the sources and influencing factors of carbon emissions, such as the impact of urbanization, industrial structure, energy consumption and land use on carbon emissions. However, most literatures select a single core variable to explore the linear relationship between this variable and carbon emissions. Compared with traditional research, this paper innovatively brings aging, agricultural mechanization and agricultural carbon emissions into a unified research system, examines their long-term equilibrium relationship, discusses the impact of aging and agricultural mechanization on carbon emissions, and demonstrates the impact and transmission mechanism of aging on agricultural carbon emissions through agricultural mechanization, as well as the dynamic relationship between them. The above is also the biggest contribution of this paper. (Line 167)
Reviewer 3 Report
It’s my pleasure to be able to review your work and suggest comments for improvement of this manuscript regarding the dynamic linkage between aging, mechanizations and carbon. It can be considered for publication in Int. J. Environ. Res. Public Health, after a major revision.
General comments:
- The novelty of this study should be clearly highlighted in the manuscript.
- All abbreviations should be explained (i.e., FAO…)
- Introduction: The authors of the manuscript should pay attention on the most important aspects related to this topic and provide a clear description of the state of art in this field.
- Methodology: The authors of the manuscript should pay attention on technical details and provide a clear description of the methodology used in this study.
- Figure 2: Authors must provide a notation for each axis of the graphs included in this figure.
- A discussion section should be added: In this section, all the results presented in the previous section should be discussed and interpreted in accordance with the main objectives of this study.
- In conclusion section is missing some perspective related to the future research work.
- Language needs to be verified throughout the document.
Author Response
First of all, we would like to thank Reviewer 3 for reading our article and for your valuable comments. Below is our response to all comments made.
Point 1: The novelty of this study should be clearly highlighted in the manuscript.
Response 1: Thank you for your suggestion. We have highlighted and emphasized the innovation and novelty of the article in line 167-176 and the last paragraph of the present article.
Point 2: All abbreviations should be explained (i.e., FAO...)
Response 2: Thank you for your suggestion. We have explained the abbreviations.
Point 3: Introduction: The authors of the manuscript should pay attention on the most important aspects related to this topic and provide a clear description of the state of art in this field.
Response 3: Thank you for your suggestion. We have made a big revision in the introduction, and made a larger elaboration on the impact of agricultural mechanization and aging on agricultural carbon emissions.
Point 4: Methodology: The authors of the manuscript should pay attention on technical details and provide a clear description of the methodology used in this study.
Response 4: Thank you for your suggestion. We have supplemented the PVAR model method in section 2.8. of the article, and the methods and models used in the article are all reflected in the article.
Point 5: Figure 2: Authors must provide a notation for each axis of the graphs included in this figure.
Response 5: We have received your suggestion, added corresponding symbols (Year/Responses) to the impulse response diagram, and explained the corresponding axes and curves at the bottom of the diagram.
Point 6: A discussion section should be added: In this section, all the results presented in the previous section should be discussed and interpreted in accordance with the main objectives of this study.
Response 6:We accept your suggestion, and the discussion part is indeed necessary. We have added it to the article (Part 4), discussing and explaining all the results presented in the previous part.
Point 7: In conclusion section is missing some perspective related to the future research work.
Response 7:I'm sorry, we missed this part, which has been added at the end of the article:“The future research can be further studied in two aspects: 1. Extending the research area and scope to the outside world, the actual situation of different countries may be dif-ferent, so it is necessary to conduct more studies in other countries, and draw more universal conclusions, so that all countries in the world can discuss coping strategies; 2. The DID model can be further used to compare the differences between the control group and the treatment group before and after the implementation of the carbon re-duction policy, so as to test the effect of the policy and improve the scientific and prac-tical effect of the policy.”
Point 8: Language needs to be verified throughout the document.
Response 8:Thank you very much for your suggestion. We have already made an inspection.
Round 2
Reviewer 2 Report
I think the authors have done a good job revising the paper according to the reviewers' comments. I am still not 100% convinced there is a direct link between Aging and Carbon Emissions. Also, why it is a dynamic link?
Author Response
Point: I think the authors have done a good job revising the paper according to the reviewers' comments. I am still not 100% convinced there is a direct link between Aging and Carbon Emissions. Also, why it is a dynamic link?
Response: Dear reviewer, thank you very much for your feedback. About the confusion, our replies are as follows:
- The question about the link between aging and carbon emissions:
(1) On the theoretical level (we cited and introduced the literature in the introduction): â‘ The physiological decline of the elderly chooses traditional primitive farming
methods or reduced farming. Studies have shown that tillage affects methane emissions, and no-till or minimal tillage is seen as a means to reduce carbon dioxide emissions and promote soil carbon sequestration, leaching or surface runoff. ②The change in population age determines the direction of land-use transition and agricultural development. Under the influence of aging and urbanization, young people migrate to cities, and problems such as shortage of agricultural labor, increasing costs, and weakening of household economic development capacity have occurred in the agricultural field, resulting in idle arable land. Farmers will adjust their land-use decisions to adapt to the impact of aging on agricultural production. Among them, one adjustment method is to abandon the land and make it non-agricultural land or let the land idle. Abandoned farmland has a positive role in mitigating climate change, and replacing agricultural cultivation with natural regeneration can better sequester carbon. The second method is to change the types of crops to be planted, choose the types of crops that are labor-saving and easy to grow, use less labor, reduce the planting scale of food crops, and increase the planting area of ​​cash crops. The last way is to rent out the land. The agricultural labor shortage caused by aging can facilitate land transactions between labor-starved and labor-surplus households. Rural land rental markets can improve land utilization by allocating land to those with high agricultural potential. ③Aging may lead to the decentralization and fragmentation of small farmers, making it difficult to form large-scale production, hindering mechanization, weakening the use of agricultural machinery, and reducing mechanical energy consumption and carbon dioxide emissions.
(2) Our empirical research results can also support our conclusions. From the regression results, both FMOLS and DOLS conclude that aging has a significant negative effect on carbon emissions. In the impulse response graph, it is also concluded that carbon emissions response to the shock of aging stabilized in a negative state, suggesting that aging can reduce agricultural carbon emissions and be effective in the long term. In summary, the conclusion that aging can reduce agricultural carbon emissions in this paper is supported by empirical evidence.
- About why they are dynamically linked?
Obviously, the carbon emissions can be caused by agricultural mechanization, mainly through energy consumption, while aging mainly affects agricultural carbon emissions through changes in labor methods and labor strategies. It can be said that mechanization is the mediating effect of aging and carbon emissions. They are dynamically linked. In addition, the impact of mechanization and aging on carbon emissions has short-term and long-term effects, which are dynamic.
Reviewer 3 Report
The authors of the paper have made a good revision based on the comments and suggestions of the reviewers.
Author Response
Dear reviewer, we have received your feedback, thank you very much for your review and affirmation of the article, we wish you every success in your work and every day!